# Cylindrical compression of thin wires by irradiation with a Joule-class short-pulse laser

Alejandro Laso Garcia [1,5], Long Yang [1,5], Victorien Bouffetier[2], Karen Appel [2], Carsten Baehtz [1], Johannes Hagemann [3], Hauke Höppner [1], Oliver Humphries [2], Thomas Kluge [1], Mikhail Mishchenko [2], Motoaki Nakatsutsumi [2], Alexander Pelka[1], Thomas R. Preston [2], Lisa Randolph [2], Ulf Zastrau [2], Thomas E. Cowan [1,4], Lingen Huang [1] ✉ & Toma Toncian [1] ✉

Equation of state measurements at Jovian or stellar conditions are currently conducted by dynamic shock compression driven by multi-kilojoule multi-beam nanosecond-duration lasers. These experiments require precise design of the target and specific tailoring of the spatial and temporal laser profiles to reach the highest pressures. At the same time, the studies are limited by the low repetition rate of the lasers. Here, we show that by the irradiation of a thin wire with single-beam Joule-class short-pulse laser, a converging cylindrical shock is generated compressing the wire material to conditions relevant to the above applications. The shockwave was observed using Phase Contrast Imaging employing a hard X-ray Free Electron Laser with unprecedented temporal and spatial sensitivity. The data collected for Cu wires is in agreement with hydrodynamic simulations of an ablative shock launched by highly impulsive and transient resistive heating of the wire surface. The subsequent cylindrical shockwave travels toward the wire axis and is predicted to reach a compression factor of 9 and pressures above 800 Mbar. Simulations for astrophysical relevant materials underline the potential of this compression technique as a new tool for high energy density studies at high repetition rates.

Dynamic shock compression serves as a crucial tool for creating warm and hot dense matter under extreme conditions that exist throughout the universe, such as the interior of planets, supernovae, and astrophysical jets. To generate these conditions in the laboratory, a wide array of techniques are employed, such as gas guns [1], pulse power systems[2,3], and nanosecond high-energy laser pulses[4,5]. Two types of shock geometries are commonly employed: planar shocks, which are prevalent in most cases, and converging shocks. Converging shocks are particularly valuable as they deliver energy to a small volume, resulting in the compression of material to exceedingly high densities and pressures. The generation of converging shocks requires precise design and facilities enabling laser irradiation with multiple beams such as OMEGA, NIF, or LMJ[6–11].

X-ray Free Electron Lasers (XFEL) provide a novel platform for studying compression and shock physics. The high number of X-ray photons per pulse, low bandwidth, short temporal pulse length and high coherence make XFELs a great tool to study ultra-fast structural dynamics via a combination of techniques: X-ray diffraction, small-

---

[1]Helmholtz-Zentrum Dresden-Rossendorf, Bautzner Landstraße 400, Dresden 01328, Germany. [2]European XFEL, Holzkoppel 4, Schenefeld 22869, Germany. [3]Deutsches Elektronen-Synchrotron DESY, Notkestraße 86, Hamburg 22607, Germany. [4]Technische Universität Dresden, Dresden 01062, Germany. [5]These authors contributed equally: Alejandro Laso Garcia, Long Yang. ✉e-mail: lingen.huang@hzdr.de; t.toncian@hzdr.de

**Fig. 1 | Experimental setup and data. a** Experimental setup of the PCI configuration used for imaging the compressed wire. The whole setup until the last 50 cm before the detector is placed in vacuum conditions, minimizing air scattering. A slit system (not shown) is used to limit the X-ray illumination to a field of view at the sample position to 250 × 250 μm and minimize fringe scattering by the CRLs (300 μm diameter). **b** X-ray PCI data measured at delays from 100 to 1000 ps after the irradiation of a 25 μm Cu wire by a 3 J, 30 fs laser pulse. The color scale gives the change in transmission compared to free-beam propagation. **c** Zoom into the red highlighted area of (**b**) for improved visibility of the converging shock.

angle and wide-angle X-ray scattering, phase contrast imaging, X-ray absorption and emission spectroscopy, etc. The combination of the XFEL beams with high-power optical laser drivers has enabled precision measurements of extreme states of matter. The generation of high-pressure states at these facilities has been restricted to the use of high-energy (60 J) nanosecond pulse duration lasers and recently upgraded to 100 J. With these drivers, scientists have been able to study the equation of state and phase transitions of materials[12–14], as well as generating conditions relevant to Earth's mantle[15], and large planet interiors[16–19]. However, the phase-space coverage is constrained in pressure range to a few Mbar due to the limited laser energy. The extension of the experimental capabilities is currently discussed by upgrade roadmaps involving coupling multi-kilojoule ns-lasers at existing XFEL instruments, like the Matter at Extreme Conditions[20] with the MEC-U upgrade at LCLS and the HED/HiBEF and with the HiBEF 2.0 upgrade at EuXFEL.

On the other hand, instruments at XFELs are also equipped with short-pulse lasers delivering Joule-level energies, pulse duration of tens of femtoseconds, and reaching intensities up to $10^{20}$ W/cm² [21,22] when focused on a sample. The interaction of such lasers with the matter generates a blast wave following the strong localized heating in the focal spot of the laser[23–28], and secondary radiation absorbed heterogeneously by the sample can drive hydrodynamic motion[29–31]. While these past experimental studies have focused on the rarefaction subsequent to the shock propagation following this blast wave, we experimentally demonstrate that by irradiating a thin wire with a short-pulse laser, conditions are met where a cylindrical shock is generated at the surface that propagates toward the wire axis. At the convergence point, this shock achieves a high compression factor and pressure. We attribute the generation of the radial compression wave to an ablative shock created by transient resistive heating of a thin surface layer of the wire. We perform particle-in-cell (PIC) simulations, shedding light on the conditions driving the shock and hydrodynamic simulations recovering the experimentally measured compression wave evolution. As an outlook we investigate the potential of this compression scheme for different materials relevant in the astrophysical context, showing that Jovian and white dwarf conditions could be reached, enabling complementary studies to those performed at kJ-class facilities.

## Results

### Experimental setup for imaging the convergent shocks

The experiment was performed at the European X-ray Free Electron Laser facility using the ReLaX laser as a relativistic plasma driver operating at the HED-HiBEF instrument[32]. A schematic of the experiment is shown in Fig. 1a. The ReLaX laser was used at 100 TW level, delivering laser pulses with an energy of 3 J on target and sub 30 fs full-width half-maximum (FWHM) pulse duration. The laser was focused employing an F/2 off-axis parabola to a spot size of approximately 4 μm FWHM resulting in an average intensity of $10^{20}$ W/cm². The 8.2 keV X-rays generated by the SASE2 undulator were used to illuminate a square region (250 μm)² around the ReLaX focal spot. This plane was imaged and magnified by a compound refractive lens[33] stack consisting of 10 beryllium lenses with a focal length of approximately 53 cm to an imaging X-ray detector located 3.3 m away. The detector was a GAGG scintillator imaged to an Andor Zyla CMOS camera via a ×7.5 objective. The detector pixel pitch is 6.5 μm, and after accounting the total magnification factor, it results in an equivalent pixel size on target of 150 nm/pixel. The imaging system was tested using resolution test targets (NTT-XRESO 50HC) to resolve 500 nm structures. It has to be noted that the resolution was limited for this experiment by the chromaticity of the SASE X-ray beam with a bandwidth of approximately 20 eV. The temporal evolution was recorded by variation of the pump-probe delay with a precision of 200 fs (RMS) given by the chosen temporal synchronization scheme (locked to the accelerator's radio frequency). The raw images are flat-fielded using the free-beam X-ray intensity distribution (without a target) and accounting for the instrument backgrounds. The main experimental results are summarized in Fig. 1b, showing Phase Contrast Images (PCI) of the wire, for different time delays ranging from 100 ps to 1 ns after laser irradiation. It is worth mentioning the used X-ray pulse duration ≤50 fs is much shorter than the typical few picosecond duration of laser-driven X-ray backlighters used conventionally in all-optical setups[34] that are also typically limited to 10s μm spatial resolution due to the source proprieties. In the PCI data, the 25 μm diameter wires are oriented vertically with the optical laser propagating from the left side and focused to the left edge of the wire in the vertical center of the illuminated area. Besides the attenuation and phase contrast generated by the wire, the evolution of two distinct structures can be measured. During the first

300 ps, a spherical shock originating from the focal spot volume is observed, similar to ref. 28. At 300 ps after the laser irradiation, this shock has already propagated through the wire. At the same time, one can follow a second nearly cylindrical shock moving radially inward toward the wire axis originating from both the left and right edges of the wire. The velocity of this shock decreases with increasing distance from the ReLaX focus. At 500–1000 ps, a convergence of the shock close to the axis of the wire can be seen. Simultaneous with the inward radial motion, a radial expansion resulting in a smoothing of the wire edge is observed. This effect is attributed to wire plasma expansion. To quantify the evolution associated with the radially converging structure, we have analyzed lineouts at 42 and 100 μm from the focal position of the laser.

We calculate the velocity of the shock by measuring the distance traveled by the shockwave between the time delays of 300 ps and 500 ps. These time delays lie within a constant shock velocity region, avoiding the initial deceleration as well as the final acceleration, as predicted by simulations and explained in the next section. An average velocity of 14.3 ± 1.3 km/s is observed 42 μm close to the laser focus and of 10.5 ± 1.3 km/s at 100 μm. Additionally, we evaluated the shock front speed emerging from the laser focus. Here, the velocity of the front decreases from 180 km/s at 20 ps to 50 km/s at 300 ps at the time of the shock release and is in agreement with ref. 28.

## Discussion
### Origin of the shocks unraveled by numerical simulations
In this section the origin of the observed cylindrical converging shocks is investigated. It is instructive to start by looking at the details of the laser-wire interaction. Particle in cell simulations are commonly used for the simulation of laser-matter interaction as predictive tools, giving detailed insights into plasma evolution at the time of laser interaction

and shortly after (at ps timescales). Performing a simulation with a wire as target, one would observe processes similar to refs. 35,36: first, the generation of a highly energetic electron population by the direct interaction of the impulsive laser pulse with the wire material that will propagate longitudinally through the wire, leading, for example, to generation of a plasma sheath with strong electric fields that accelerate ions[37]. Simultaneously, part of the hot electron population will move at close to the speed of light transversely away from the focal spot. The hot electrons are electrostatically trapped close to the wire surface, and in their wake, the wire surface is ionized. To achieve charge balance a return current along the surface is established reaching current densities of $10^{13}$ A/cm². These currents encompass the whole wire surface up to a skin depth. Proton imaging techniques (employing laser-accelerated protons) have been successfully applied to measuring the sheath fields and thus the dynamics of the associated hot electron transport and subsequent return current[35,38–40].

While the lifetime of such currents was estimated by Quinn et al. to be 20 ps[35], one order of magnitude longer than the ps duration of the laser pulse used, recent theoretical work has been focused on laser pulses matched to our experiment of several tens fs[41], work that follows a multi-scale approach. PIC simulations are used in the first step to calculate the current density profile along an extended wire. With a model detailed by equation (1), a temperature distribution is evaluated from the current density within the skin depth at the wire surface. Finally, this temperature distribution is used as initial conditions for hydrodynamic simulations to predict the long-term shock formation, propagation and density compression. To confirm this scenario, we have performed 2D PIC simulations of the laser interaction with a 10 μm diameter Cu wire target. We observe the formation of a return peak current of $2.8 \times 10^{13}$ A/cm², with a lifetime of 100fs (Fig. 2a shows $j_y$ and associated magnetic field for $t = 38$ fs after the peak of the

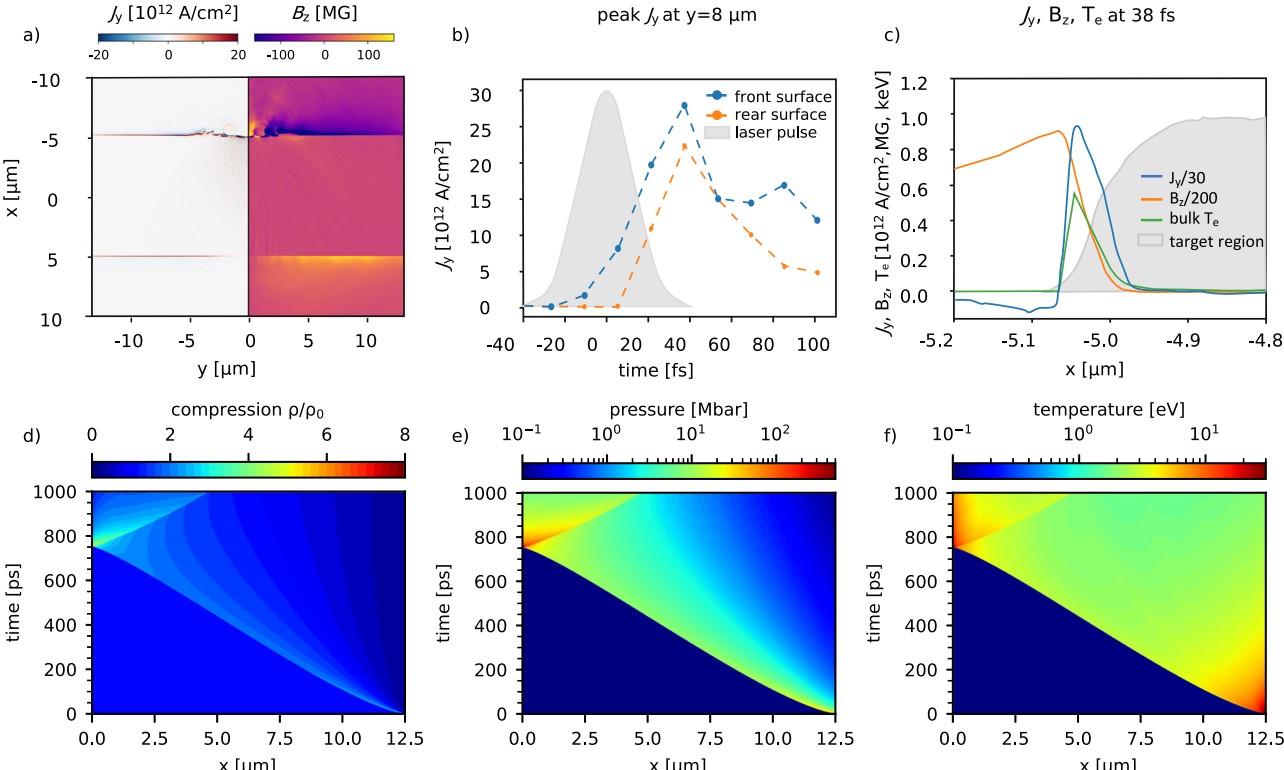

**Fig. 2 | Shock formation and propagation simulations.** a, b and c correspond to the 2D PIC simulations using a copper 10 μm diameter wire; d, e and f correspond to the 1D hydro simulations using the SESAME equation of state. a Snapshot of the current density parallel to the wire axis and the associated magnetic field strength 38 fs after peak of the interaction, b time evolution of peak current density 8 μm

away from the laser focus and c distribution of the current density, magnetic field, bulk temperature for the surface layer for $t = 38$ fs. Hydrodynamic temporal evolution of compression factor (**d**), pressure (**e**) and temperature (**f**) for a shock driven by a surface temperature of 250 eV.

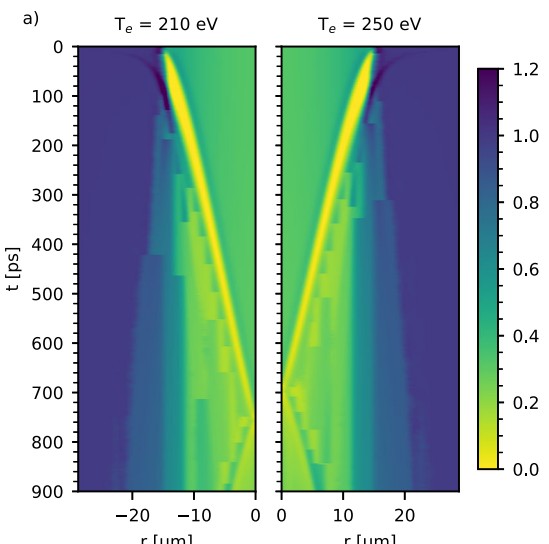

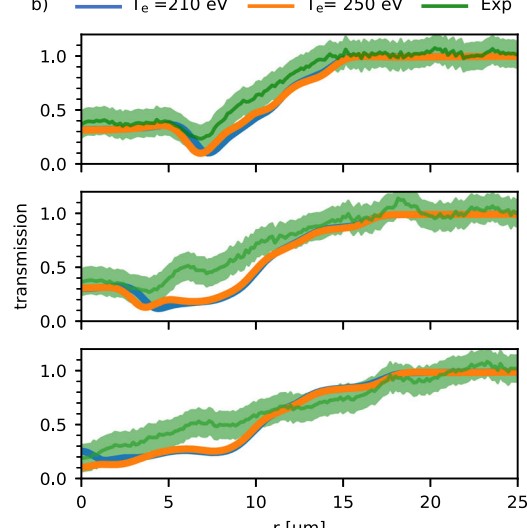

**Fig. 3 | Results of synthetic PCI calculation and comparison with experimental data. a** Simulated PCI profiles using the hydrodynamic simulation data for initial temperatures of 210 and 250 eV. **b** Comparison of experimental and simulated data 300 ps, 500 ps, and 700 ps for 210 eV and 250 eV initial temperature. The matching of the experimentally observed convergence at 700 ps occurs for the 250 eV case.

interaction, and Fig. 2b the time evolution peak of $j_y$ at a distance of 8 μm away from the focus). The surface return current has two effects on the wire target which can lead to compression: the magnetic compression and Joule heating and the associated ablation. While the return current magnitude scales inverse of the wire radius[41], for a 25 μm diameter copper wire, the peak of the surface return current density is predicted to be in the range of $0.4$–$1.1 \times 10^{13}$ A/cm², with a current strength decaying further away from the laser focus. At the time when the return current is maximal, the thermal pressure is evaluated to be 8 times higher than the magnetic pressure of 250 Mbar. The resulting plasma $\beta \approx 8$ confirms the kinetic nature of our shock formation. The electron temperature distribution to be used as initial condition for the hydro simulations is estimated using the electron energy equation[41,42],

$$\frac{3}{2} n_e \frac{\partial T_e}{\partial t} = \frac{\partial}{\partial r} \left( K_{T_e} \frac{\partial T_e}{\partial r} \right) + \frac{j_e(r)^2}{\sigma_{T_e}}, \quad (1)$$

where $K_{T_e}$ is the thermal conductivity of cold electrons, $\sigma_{T_e}$ is the electric conductivity, and $j_e(r)$ is the surface return current distribution in the radial direction from PIC (Fig. 2c). The equation's right side accounts for heat diffusion in the radial direction and the Joule heating due to the surface return current. The electron resistivity model and the heat diffusion coefficient are functions of temperature and density and can be extracted from the SESAME equation of state[43]. The calculated electron temperature with this equation peaks at 140–320 eV for the return current range of $0.4$–$1.1 \times 10^{13}$ A/cm² and is distributed within a 0.1 μm skin depth layer. This temperature distribution is used as the initial condition for hydrodynamic simulation to investigate plasma evolution until 1 ns. These simulations with the FLASH code[44,45] solve in a 1D cylindrical symmetry the one fluid, two-species (ion and electron) and two-temperatures hydrodynamic equations with copper SESAME equation of state[43]. The ion temperature and electron temperature are assumed to be equal due to the high collision rates between these.

Figure 2d–f shows the result of the simulation of the shock dynamic for a 25 μm copper wire under the experimental laser irradiation conditions, giving the temporal and spatial evolution of compression (density normalized to initial density), pressure and temperature. The initial condition used is a peak temperature of 250 eV with an exponential decay depth and a decay constant $\tau = 0.067$ μm.

The shock is formed within the first 5 ps due to the ablation pressure. It starts at the surface with a compression factor of 2.8 with respect to cold copper and a peak pressure of 111 Mbar. It travels toward the wire axis with a starting velocity of 44 km/s. At a time of 200 ps, the shockwave has decelerated to 15 km/s, and the temperature of the shock front reaches 22 eV, while the pressure decreases to 12 Mbar. The peak compression factor at this point is 2. Between 200 ps and 650 ps, the shock propagates with constant conditions, and the shock front moves from 8.1 μm away from the wire axis down to 1.1 μm. After this point, the shock gains velocity until it converges at the wire axis, reaching compression factors of 9, corresponding to densities of 80.6 g/cm³, a temperature of 38 eV and a pressure of 830 Mbar.

**Comparison between experimental and synthetic imaging data**
Using the radial density profile from the hydrodynamic simulations, the expected X-ray PCI profiles were calculated and compared with experimental results. Using a forward Abel transform of the density profile, the projected mass density is obtained as probed by the X-ray in the experimental geometry. The intensity at the detector plane can be calculated via the Transport of Intensity Equation[46], as:

$$I(x_1, z = z_1) = I(x, z = 0) \left( 1 + \frac{z_1}{k} \nabla^2 \Phi(x, z = 0) \right)^{-1} \quad (2)$$

where $I(x_1, z = z_1)$ is the intensity at the detector located at a propagation distance $z_1$, $I(x, z = 0)$ is the intensity at contact, that is directly at the exit plane of the target and given by the attenuation of the X-rays by the target, $k$ is the X-ray wave-vector, and $\Phi(x, z = 0)$ is the phase shift at contact. The PCI data measured with an undriven wire was used to characterize the propagation distance, resulting in an equivalent plane located at a distance $z_1 = 6$ mm after the target that is imaged by the CRLs onto the detector. Figure 3a shows the temporal evolution of the forward calculated PCI pattern up to 1 ns for two initial temperatures, 210 eV and 250 eV. The convergence time is 758 ps for the 210 eV simulation and 698 ps for 250 eV. The resulting profiles at time-steps 300, 500 and 700 ps are compared in Fig. 3b with experimental data 42 μm away from the laser focus. The synthetic profiles reproduce quantitatively and qualitatively the experimentally observed PCI patterns with features such as the position of the inward propagating shock-front and outward beam refraction. In this context,

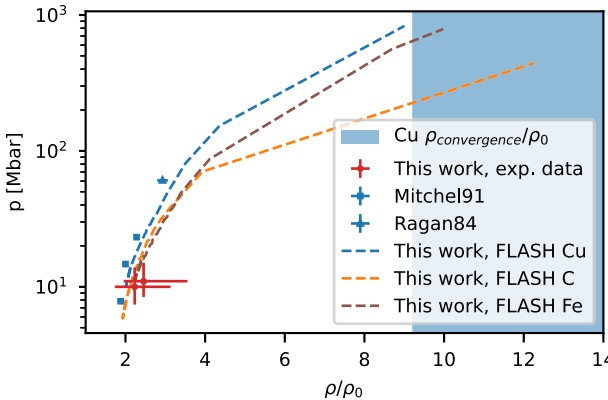

**Fig. 4 | Shock states for different materials.** The red dots represent the pressure extracted at delays of 300 ps and 500 ps for copper by this work, the squares and triangle corresponds to previous published results[58,59]. The blue area represents the experimental compression factor at convergence within experimental uncertainties. The blue dashed line is states reached for Cu according to hydrodynamic simulations, and the brown and orange lines are Fe and C simulations.

it is important to consider the effect of temperature-induced opacity changes. Using the TOPS/ATOMIC database[47], it was confirmed that the values for the mass attenuation coefficient for a temperature of 22 eV differ on percent level from those of cold copper material, thus it can be neglected for the rest of the analysis.

We have selected the experimental data at 300 ps and 500 ps delay and further analyzed the PCI profiles at 42 μm away from the laser focus to extract the shock parameters. As the shock acceleration is minimal between these delays, the uncertainty in the velocity estimation and its effect on the shock pressure is reduced. The generalized Paganin method[48] was applied to calculate the intensity at the target plane. Here the assumption is that all the intensity variation is due to absorption in the wire. An inverse Abel transform is used to extract the radial mass attenuation and, consequently, the density profile. Finally, the Rankine-Hugoniot equation is used to calculate the shock pressure using the experimental shock velocity and density, obtaining a value $p = 11.0^{+4.0}_{-2.6}$ Mbar for 300 ps and $p = 10.0^{+3.9}_{-2.6}$ Mbar for 500 ps. The values extracted from hydrodynamic simulations are 10.5 Mbar and 10.6 Mbar, in close agreement with the experimental value. Since an experimental velocity was not available, only the evaluated central density of $104^{+21}_{-21}$ g/cm³ for convergence at 700 ps is highlighted in Fig. 4.

### Scaling for various materials

A comparison of this value with previously published results for copper is shown in Fig. 4, together with the predicted states achieved in our experiment according to the FLASH simulations. Furthermore, Fig. 4 shows simulation predictions for carbon and iron as representative materials in the context of astrophysical research. Using the same return current conditions and target diameter as for the Cu wires, the simulations predict pressures up to 790 Mbar for iron and up to 400 Mbar for carbon. The temperatures, in these cases, range up to 32 eV and up to 16 eV for iron and carbon, respectively, at the time of convergence. The carbon states are comparable to the ones expected in Jovian worlds as well as exoplanets[49], showing the potential of this platform for planetary interior research. The iron states are in the range of the stellar conditions for white dwarf envelopes, as shown in MJ experiments at NIF[50]. While further investigation is beyond the scope of this paper, the consideration of higher dimensionality of the compression is paramount for a fully quantitative prediction of the compression capabilities. First 2D simulations assuming a 33% drop of the initial temperature between the front and back surface of the wire show that the maximal density would decrease by 25% compared to

the case of ideal compression, demonstrating the robustness of the process and potential of this method as a platform for HED studies.

In summary, we have demonstrated the capabilities of a Joule-class laser irradiating thin wire targets to generate extreme pressure states relevant to astrophysical studies. We have shown how the state can be characterized via imaging techniques exploiting the ultra-short duration and high brilliance of an XFEL beam. In particular, converging cylindrical shocks in copper with pressures up to 11 Mbar have been measured, with simulations predicting pressures up to 830 Mbar at convergence, supported by the excellent quantitative agreement between experimental and forward calculated data. This method of shock-generation paves the way for performing astrophysical experiments in the laboratory providing large statistics thanks to the high repetition rate of the lasers (shot per minute) and involving simple and ubiquitous targets.

## Methods

### X-ray setup

The X-ray beam was characterized in energy via the elastic scattering of the beam on a YAG scintillator. The elastic signal was measured via a von Hamos X-ray spectrometer[51], which was previously calibrated via copper $K_\alpha$ emission. The energy was determined to be 8.2 keV. The pulse energy of 600 μJ on average was measured via an X-ray gas monitor (XGM) in the X-ray tunnel. An X-ray lens configuration was chosen that resulted in a pencil-like beam at the target chamber (vacuum in the $10^{-5}$ mbar range). The beam size was measured with a YAG scintillator at the pulse arrival monitor located 9.5 m before the target chamber center (TCC) and at 3.3 m after TCC with the same detector used for PCI measurements. The beam size at TCC was interpolated between those two points. The compound refractive lenses stack consisted of 10 Be lenses, with a radius of curvature of 50 μm manufactured by RXOptics Germany with a web thickness of 50 μm for each lens. The resolution of the system was characterized by imaging of a Siemens star test target, NTT-XRESO-50HC. The resolution target is made of tantalum with a thickness of 500 nm. The imaging of the target is shown in Supplementary Fig. S1.

### Optical laser setup

The ReLaX laser was used at 100 TW energy level, delivering 3 J of energy on target. The pulse duration was optimized using a self-referencing spectral interferometer WIZZLER 800 by Fastlite and was regularly checked for best compression by a second harmonic generator autocorrelator. The pulse intensity contrast was measured to be within the specs presented by Laso Garcia[21]. The focal spot quality was monitored and optimized by using a ×20 APO PLAN microscope objective. The spatial phase was optimized by an adaptive deformable mirror coupled to a wavefront sensor. The synchronization between optical and X-ray laser was measured by spatial photon arrival monitor techniques[52].

### Data processing

Each of the X-ray images taken was flat-fielded according to the following procedure: first, the detector background is subtracted by subtracting the average of 313 empty frames. Then the image is normalized by the pulse energy measured by the XGM. Next, the scattering pattern generated by the slits on the shot of interest is compared to an ensemble of scattering patterns taken without target. The slit scattering pattern is sensitive to the X-ray intensity and the beam pointing. A chi-square minimization is used to find the best match. The normalized on-shot image is divided by the normalized free-beam best match. Finally, any residual intensity variation due to imperfect match is locally corrected by fitting a third-order polynomial to an area 200 pixels around the wire shadow and dividing by the fit result (shown in Supplementary Fig. S2). The final uncertainty of the flatfield is obtained from the peak-to-valley transmission variation

associated with the transmission baseline. This results in an uncertainty of $s(T) = 0.12$.

## Particle in cell simulations

The 2D PIC simulations are performed with the PICLS code[53] to evaluate the surface return current. To best match the experimental conditions, we first simulate the interaction of the prepulse of ReLaX laser and Cu targets using the MHD code FLASH, which gives the initial density profile for the PIC simulations. The main pulse of the ReLaX laser is modeled by a Gaussian profile both in the spatial and temporal dimensions with full-width at half-maximum (FWHM) spot size $w_{FWHM}$ = 4 μm and duration $\tau_{FWHM}$ = 30 fs respectively, resulting in a peak intensity $I_0 = 5 \times 10^{20}$ W/cm$^2$. In order to resolve the plasma wavelength for a fully ionized Cu plasma, that corresponds to an electron density of 1400 $n_c$, where $n_c = 1.74 \times 10^{21}$ cm$^{-3}$ is the plasma critical density, the cell size and time step are set $\Delta x = \Delta y = 5.3$ nm and $\Delta t = 0.0178$ fs respectively. The simulation box consists of $N_x \times N_y = 7500 \times 5000$ cells, corresponding to the real space size of $40 \times 27$ μm. The PIC simulations use an absorbing boundary condition and include field and direct impact ionization models[54]. In addition, the relativistic binary collisions between charged particles are included. To save computational time, the diameter of the Cu wire is reduced from 25 μm to 10 μm while maintaining the same level of fast electron refluxing within the target[55].

## Hydrodynamic simulations

The FLASH code as version 4.6.2, developed by the University of Rochester, was employed for hydrodynamic simulations. These simulations utilized a 1D cylindrical symmetry geometry. The total simulation box is 50 μm in size, with the wire target material occupying 12.5 μm and the rest containing vacuum. This vacuum is filled with low-density hydrogen at a density of $10^{-5}$ gcm$^{-2}$ and a temperature of 1 eV. The target material varies depending on the case and includes copper, iron, or carbon. Each target region contains solid targets at their respective solid densities. The initial temperature distribution of the target is determined using the electron energy equation 1. Based on the return current scaling theory, magnetic compression driven by the $\mathbf{J} \times \mathbf{B}$ force is disregarded, setting the initial fluid velocity to zero. The boundary conditions use a reflective boundary condition for the symmetry axis and free space for the vacuum. A self-adaptive mesh grid and derived material properties from the corresponding SESAME equation of state were used.

## Synthetic PCI data

The density output of the hydrodynamic simulations is used to calculate the synthetic PCI intensity $I(x, z)$ with x the transverse distance from the axis of the wire relative to the X-ray propagation direction and z the distance from the wire along the X-ray propagation distance. In the first step, the Abel transform is evaluated

$$\Gamma(x) = 2 \int_x^\infty \rho(r) r / \sqrt{r^2 - x^2} dr, \quad (3)$$

to obtain $\Gamma$ the total mass density projected along the line of sight $x$ using $\rho(r)$ the 1D mass density in cylindrical coordinates. From $\Gamma(x)$, the absorption mass attenuation coefficient $\mu(x)$ and the phase $\Phi(x, z = 0)$ are calculated at the exit plane of the wire. The change of PCI intensity after a propagation distance $z$ is given by

$$I(x, z) = I_0 e^{-\mu(x)} \left(1 + \frac{z}{k} \nabla^2 \Phi(x, z = 0)\right)^{-1}, \quad (4)$$

with $I_0$ the original intensity, $k$ the wave number of the X-ray. In the last step, we consider both the experimental resolution and also the

bandwidth of the SASE X-ray beam. The limitations of the resolution are simulated by applying a Butterworth filter with a cut-off frequency of 0.00057 nm$^{-1}$ corresponding to the Nyquist frequency of the detector imaging system. The SASE bandwidth effect is calculated by approximating the spectral distribution as a Gaussian with 20 eV FWHM and sampling 40 wavelengths $\omega$. For each of these wavelengths, a corrected $z$ (compared to the 6 mm) is used to obtain $I(x, z(\omega))$. The final PCI intensity is integrated from the $I(\omega)$ with weights given by the spectral distribution.

## Shock density reconstruction

The procedure to reconstruct the density from experimental PCI profiles is the direct inversion of the procedure to produce the synthetic PCI data. The Paganin method is used to invert the TIE equation and obtain the intensity profile at target contact. The intensity at contact, $I(x, y, z = 0)$ relates to the measured one, $I(x, y, z)$ as

$$I(x, y, z = 0) = -\log_e \left( \mathcal{F}^{-1} \frac{\mathcal{F}[I(x,y,z)/I_0]}{1 - 2z \frac{\delta}{\mu d^2}(\cos(dk_x) + \cos(dk_y) - 2)} \right) \quad (5)$$

where μ is the mass attenuation coefficient, $\delta$ is the real part of the index of refraction, $d$ is the pixel size on target and $k_{x,y}$ are the spatial frequencies at the detector plane, and $\mathcal{F}$ the Fourier transform. The intensity at contact is then directly related to the mass attenuation coefficient. The attenuation as a function of radius is calculated via the inverse Abel transform of the intensity. Finally, division by the mass attenuation coefficient returns the radial density profile. An estimate of the uncertainty of the Abel reconstruction was obtained by shifting the center of rotation by a distance of 500 nm around the nominal axis to account for the finite resolution. This procedure was done for the experimental intensity profile as well as the upper and lower boundaries. The final peak density and uncertainty are calculated as the average and standard deviation of the peak density of all profiles. The peak density for the 300 ps data point is $\rho = 22^{+10}_{-5.0}$ gcm$^{-3}$, for the 500 ps $\rho = 20^{+8.4}_{-4.5}$ gcm$^{-3}$ and for 700 ps $\rho = 104^{+21}_{-21}$ gcm$^{-3}$.

## Calculation of the shock pressure

The pressure was extracted from the experimental data via the Rankine-Hugoniot relation. The pressure can be expressed as:

$$p = p_0 + u_s^2 \rho_0 \left(1 - \frac{\rho_0}{\rho_s}\right) \quad (6)$$

where $p$ is the shock pressure, $p_0$ is the pressure of the matter in front of the shock, $u_s$ is the shock velocity, $\rho_0$ is the uncompressed density and $\rho$ is the compressed density. In our case, the pressure in front of the shock is ambient pressure. The shock velocity can be extracted from the imaging data by measuring the distance traveled by the shock between two time delays, specifically between 300 ps and 500 ps. The uncertainty in the timing measurement is negligible due to the synchronization of the beams with an RMS <200 fs. The uncertainty in the position is limited by the imaging resolution of ≈500 nm. With this approach, the shock velocity results in $u_s$ = 14.3 ± 1.3 km/s (at 42 μm from the interaction point). The density reconstruction has been discussed in the previous section. The uncertainty on the pressure is calculated by propagating the uncertainties of the reconstructed density and shock velocity.

## Data availability

Data recorded for the experiment at the European XFEL are available at EuXFEL data repository, HED 4597[56], after the expiration of the embargo period or upon reasonable request. The simulation data used to generate Figs 2–4 are available at the Rossendorf data repository[57].

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

## Acknowledgements

We acknowledge the European XFEL in Schenefeld, Germany, for the provision of X-ray free electron laser beam time at the Scientific Instrument HED (High Energy Density Science) and would like to thank the staff for their assistance. The authors are indebted to the HIBEF user consortium for the provision of instrumentation and staff that enabled this experiment. FLASH was developed in part by the DOE NNSA and DOE Office of Science-supported Flash Center for Computational Science at the University of Chicago and the University of Rochester.

## Author contributions

A.L.G. and T.T. conceptualized the experiment. L.Y. and L.H. developed the theory and performed the PIC and hydrodynamic simulations. L.Y., L.H., A.L.G. and T.T. analyzed the simulations. A.L.G., V.B., K.A., C.B., H.H., O.H., M.M., M.N., A.P., T.R.P., L.R. and T.T. performed the experiment. A.L.G., L.Y., L.H., T.E.C., J.H. and T.T. analyzed the data. A.L.G., L.Y., L.H., T.E.C. and T.T. wrote the original manuscript draft. All authors, including T.K. and U.Z., reviewed and edited the manuscript. L.H. and T.T. supervised the project.

## Funding

## Competing interests

The authors declare no competing interests.
