## [Peer Review File · Nature Communications]

Cylindrical compression of thin wires by irradiation with a Joule-class short pulse laserEditorial Note: Parts of this Peer Review File have been redacted as indicated to maintain the confidentiality of unpublished data.

REVIEWER COMMENTS

Reviewer #1 (Remarks to the Author):

This study uses x-rays from a free electron laser to examine the evolution of a 25 μm diameter Cu wire with phase contrast imaging. The authors claim that the data shows the propagation of a strong implosive shock wave driven by return current heating of the target surface. Radiation hydrodynamics simulations using the FLASH code are presented to illustrate that some of the behavior seen matches the behavior to be expected of an imploding shock wave.

In order for the conclusions presented to be convincing, it is my opinion that this study requires both additional experimental measurements and also a more rigorous attempt at modelling the experiment. In addition the work needs to be properly set within the framework of the information that already exists in the literature. There are numerous previous studies, going back more than 15 years, discussing the driving of shock waves with short pulse lasers - both employing high energy systems like Vulcan PW and Titan as well as smaller Joule class lasers. There are also quite a lot of previous studies of short pulse laser interaction with wire targets, including Cu wire targets, of similar (even identical) diameter to that being used here. The authors awareness of these studies seems quite limited, based on the work that has been cited. While it is understood that the laser here is a relatively low energy short pulse laser, even the investigations related to higher energy systems should be being cited and discussed more extensively, since much of the physics would be expected to be similar (regards driving hydro), even if the relevance for some applications could be diminished by scale/cost/rep-rate. However, the fact that, for instance, the studies that have been carried out by the TIFR group and collaborators are not cited is concerning, as they have performed and published multiple studies of shock waves launched by (what seems like) an almost identical laser to that employed here. Furthermore, the simulation methodology used in some of these studies is more robust, using a sequence of simulations to attempt to model, prepulse, main pulse and post-pulse evolution of the target using a rad-hydro > PIC > hydro framework.

The data presented may show what the authors suggest, but I do not find this single diagnostic output sufficiently compelling, taken alone and without more robust simulation support. High resolution X-UV imaging, x-ray spectroscopy, and other diagnostics, would be well suited to measuring the conditions produced by resistive heating (and possibly the emission from shock waves if the diagnostic is streaked). Such diagnostics have been fielded successfully on previous experiments (including of Cu wire targets of similar dimensions to those used here) by collaborations involving LLNL, OSU, UCSD and others. The single diagnostic approach taken is inadequate to support the claims being made by the authors. There are features in the radiographs that are not well explained by their hypothesis- for instance at 500ps there appears to be a doubling of the brighter structure on the RHS of the image. If we are to believe that such features are always shock waves, then where does the second shock on the RHS come from? A sudden surface heating should produce just one shock (visible on either side of the target), and a simple homogenous cylinder offers no boundary that could lead to break-up of the primary shock. If such features are not necessarily shock waves, how can we be confident that other features have been identified correctly as shock waves? It is interesting data, but in my view substantially more work is needed - both on the experimental and simulation fronts, to make a compelling case for publication. Even then though, the idea of launching shock waves with short pulse lasers is not new - even in a cylindrical geometry. Indeed similar schemes have been proposed previously (for instance by Y. Sentoku) as a potential source of neutrons in heterogenous cylindrical targets imploded by surface resistive heating.

As regards applications - a convergent geometry is wholly unsuitable for making precise measurements of EOS. It does not offer the necessary diagnostic access and shock wave propagation is complicated by the convergent geometry. EOS experiments are almost always performed in planar geometry with a drive that is as spatially and temporally uniform as possible. The fact that you can launch a strong shock by some means in no way implies that you have developed a suitable basis for

doing EOS experiments.

In summary I find the paper unconvincing and I am concerned by how limited the author's awareness seems to be of related previous studies. I also find the suggestion that such a platform could be used for EOS studies is poorly considered. I do not recommend this paper for publication.

Reviewer #2 (Remarks to the Author):

Review Report on manuscript NCOMMS-24-04117-T

"Cylindrical compression of thin wires by irradiation with a Joule-class short pulse laser" by Alejandro Laso Garcia, Long Yang et al.

The manuscript presents experimental evidence of dynamic convergent shock compression in thin metallic wires driven by the interaction of tens-of-femtoseconds-duration, Joule-energy level, relativistic-intensity single laser pulse. The results were obtained by adding the radiography probing capability of an XFEL 8.2 keV x-ray source, producing phase contrast images (PCI) of the driven wires at different times.

While this is not the first work providing evidence that short-pulse intense laser pulses can launch strong shocks in solid-density targets [e.g. Budil et al., *Astrophys. J. Suppl. Ser.* 127 262–5 (2000) (a missing seminal reference, using a low-repetition-rate PW laser system); Santos et al., *New J. Phys.* 19 (2017) 103005 (should be mentioned that they also used a high-repetition -rate Joule-level laser driver)] – and, as such, works as an interesting extension to ns-laser drivers, not yet broadly explored capability to creating warm and hot dense matter states of relevance for studies in planetology and/or astrophysics – the results presented here are unprecedented: For the first time it is clear shown such shock propagation inside the dense targets, and this with excellent temporal and spatial resolutions that were unreachable in previous studies using exclusively optic-laser sources.

The formation of the converging shock by a transient energy deposition over a sub- μm thin layer all-around the wires, and this at tens of μm height-offset in respect to the interaction laser axis, is convincing as the experimental features seen in the phase contrast images (shock velocity, transmission radial profiles at selected times) are fairly well reproduced by post-processed 1D cylindrical symmetric hydrodynamic simulations.

The data obtained is of interest and useful for specialists in the fields of high-power laser interaction with matter and extreme states of matter, and – as claimed by authors – does have the potential for studies relevant to conditions found in Jovian planets or in stars, that is a basis of knowledge development in the broad field of high energy density physics.

This said, in my opinion, the present manuscript does not sufficiently detail quantitatively the transient, kinetic processes by which the high-intensity laser generates a surface current of hot electrons around the cylinder and the opposite, quasi-instantaneously generated, return current of thermal electrons, and that it is the energy deposition by the resistive current transport that drives the ablation of the laser surface and subsequent shock formation. The process is sufficiently described qualitatively providing the needed references from the vast literature on the subject, but the numbers (surface current densities of 10^{13} A/cm², surface heating to electron temperatures above 300 eV) come from unpublished PIC simulation material (cited arXiv reference [36]) that does not study the Cu wires used in this experiment and its exact configuration.

It is the opinion of this referee that the presented PCI data and the physics that it unveils are of the highest standards and deserved to be published, but to meet the publication standards of *Nature Communications*, the manuscript should provide a comprehensive description of the full system, encompassing both its kinetic and hydrodynamic aspects. In other words, the referee suggests that

the authors add results from PIC simulations reproducing the interaction of the high-intensity laser with a 25- μm -diameter Cu wire, and demonstrate the range of reached surface currents, magnetic-field and transient heating of the material that, ultimately, can allow to discuss on more solid grounds the energy source for driving the observed dynamic radial-shock compression.

Minor questions:

- Shouldn't the colorbar scale for transmission in Fig. 1 b) be limited to a maximum of value of 1?
- Why isn't the transmission profile data also analysed at 700 ps, when the maximum compression is reached? Why the corresponding data does not appear if the plot of Fig. 4? This would give a more robust ground to claim pressures in the range of Gbar...

Other minor suggestions:

- Adding the spatial scale to Fig. 1 c).
- In the literature, the symbol j_h is commonly used for the direct laser-accelerated current-density of hot electrons, not for the return current of thermal electrons for which it is commonly preferred j_e or j_r .

Cylindrical compression of thin wires by irradiation with a Joule-class short pulse laser

Reviewer 1 (Remarks to the Author)

This study uses x-rays from a free electron laser to examine the evolution of a 25 um diameter Cu wire with phase contrast imaging. The authors claim that the data shows the propagation of a strong implosive shock wave driven by return current heating of the target surface. Radiation hydrodynamics simulations using the FLASH code are presented to illustrate that some of the behavior seen matches the behavior to be expected of an imploding shock wave.

In order for the conclusions presented to be convincing, it is my opinion that this study requires both additional experimental measurements and also a more rigorous attempt at modelling the experiment.

In addition the work needs to be properly set within the framework of the information that already exists in the literature. There are numerous previous studies, going back more than 15 years, discussing the driving of shock waves with short pulse lasers both employing high energy systems like Vulcan PW and Titan as well as smaller Joule class lasers. There are also quite a lot of previous studies of short pulse laser interaction with wire targets, including Cu wire targets, of similar (even identical) diameter to that being used here. The authors awareness of these studies seems quite limited, based on the work that has been cited. While it is understood that the laser here is a relatively low energy short pulse laser, even the investigations related to higher energy systems should be being cited and discussed more extensively, since much of the physics would be expected to be similar (regards driving hydro), even if the relevance for some applications could be diminished by scale/cost/rep-rate. However, the fact that, for instance, the studies that have been carried out by the TIFR group and collaborators are not cited is concerning, as they have performed and published multiple studies of shock waves launched by (what seems like) an almost identical laser to that employed here.

We fully acknowledge the referee comments that the generation of shocks by short pulse lasers has a long standing history. Our intentions were not to disregard previous work, but while these direct laser-driven shocks are ubiquitous, the mechanism driving the initial condition, that we have determined to be the source of our observed cylindrically imploding shock, differs from those previously reported. In the past, three different shock wave mechanisms have been proposed, theoretically and experimentally studied, in the context of impulsive ps-fs short pulse laser matter interaction:

1. The shock wave generation and modelling focusing on the shock/blast wave that develops from the laser focus.

This mechanism has been extensively studied previously (restricting ourselves just to experimental literature), and is also observed in our experiment; however, it is not the focus of our investigation. Here, the origin of the shock is the highly localized, transient heating due to direct laser-plasma interaction that results in a hot spot with temperatures often reaching several keV, driving a hydrodynamic shock due to the high temperature imbalance with the ambient bulk on a ps timescale. There are indeed a multitude of studies, some examples are listed in following starting with (Budil 2000) as a seminal paper, (Akli 2008) where K-shell emission x-ray spectroscopy was used to investigate the localization of heating subsequent shock, (Mondal 2010) where the dynamic of the shock wave was explored by Doppler-shift of a probe beam, (Lancaster 2017) that used XUV imaging to observe the propagation of the shock through a foil target. While the studies listed above used PW lasers such as NOVA or VULCAN, there are studies employing J-class lasers similar to our experiment such as those from the TIFR group (Kamalesh 2018) where the change of reflectivity was used to reconstruct the breakout history of the shock, or (Santos 2017) where streaked optical pyrometry was used to characterize the shock wave.

[Redacted]

2. Shock wave generation by isochoric heating in a heterogeneous solid.

Sentoku et al. (2007) proposes to use a layered slab target, where layers can be isochorically heated differently (CD-Al-CD). As the layers are heterogeneously heated, a pressure gradient and subsequent shock wave is formed that sustains density and temperatures that would lead to a higher thermal neutron yield compared to plain targets. Opposite to our cylindrical shock wave leading to a compression on axis, the setup used by Sentoku in this study generates radial outwards directed shocks. One could argue that by inverting the layer geometry a cylindrical shock propagating towards the laser axis could be formed in this way. As the formation of the shocks is dependent on efficient isochoric heating by 100 keV electrons having a limited divergence, the region of the target that will allow such shock formation will overlap with the shock wave discussed at point 1 that propagates faster and will engulf it, for the typical diameter of the target we report.

3. Usage of heterogeneous targets with varying resistivity in the context of strong current transport in wires associated to hot electron transport and collimation as hydrodynamic driver.

Robinson et al. (2013) have proposed to use a hot electron transport in a wire with a radial graded resistivity to heat up the wire heterogeneously and use it as hydrodynamic driver. The heating (electron) itself is homogeneous, but the electron density is heterogeneous -- depends on Z of layer -- and hence there is a strong pressure gradient ($nk_B T$) at the interface due to the local electron density. While the heating is discussed at length, the subsequent hydrodynamic evolution is not discussed, thus it is not clear how the quality of the compression would be. It's worth mentioning, that extensive literature can be found where strong current transport has been investigated both experimentally and in theory in the context of fast ignition. Here in particular, studies using cone attached wires (Kodama 2004) are of interest. The wire facilitates collimation of the electron beam that would quickly diverge otherwise. The heating of the wire induced by the interaction with the strong current is fundamental different, leading to

bulk heating and a hydrodynamic driven by a magnetized plasma. Green et al. (2007) report on optical probing measurements where a non-homogenous heating of a cone-attached wire is reported. The dynamic of the over-dense plasma region (inaccessible to optical probing) is compared with a series of hydrodynamic simulations with varying initial condition. The authors find as best matching scenario a wire with a high localized heating at the surface (due to a strong return current) 4x higher than the core temperature of 100eV (heated directly by the hot electron beam). The impact of such a configuration for generation of a converging shock wave is not discussed, just the late time expansion of the plasma is compared to the experimental results.

Changes in the manuscript: We have included more references (Budil, Modal, Akli, Lancaster, Jana, Santos, Sentoku, Green, Robinson) to previous experimental investigations of shocks subsequent to short pulse laser matter interaction. Also we have strengthened the difference of the physical process driving the shocks observed in this investigation compared to previously reported, the corresponding paragraph reads now:

The interaction of such lasers with the matter generates a blast wave following the strong localized heating in the focal spot of the laser [Budil, Modal, Akli, Lancaster, Jana, Santos], and secondary radiation absorbed heterogeneously by the sample can drive hydrodynamic motion [Sentoku, Green, Robinson]. While these past experimental studies have focused in the rarefaction subsequent to the shock propagation following this blast wave, we experimentally demonstrate that by irradiating a thin wire with a short-pulse laser, conditions are met where a cylindrical shock is generated at the surface that propagates towards the wire axis.

Furthermore, the simulation methodology used in some of these studies is more robust, using a sequence of simulations to attempt to model, prepulse, main pulse and post-pulse evolution of the target using a rad-hydro > PIC > hydro framework.

For the intensity contrast of the laser ReLaX (10^{-10} @1 ns, 10^{-10} @100 ps, 10^{-8} @ 50 ps, 10^{-7} @ 5 ps) we do not expect any significant pre plasma formation besides a very localized heating few μm around the focal spot of 4 μm FWHM. At tens to hundreds of μm away no pre plasma formation is expected. This is consistent with experimental observations, thus starting with a PIC simulation is a valid approach. The transfer of the PIC extracted parameters into initial conditions of the hydro code, is detailed by the paper (Yang 2024). Nevertheless, as also requested by the second referee, and to mimic closer realistic initial conditions as input for the hydro simulation we have conducted and added to the manuscript the results of PIC simulations, limited due to computational restrictions to a 10 μm wire when using Cu as target material. The prepulse effect on the target is simulated as shown in Fig 2 of the response. The results of this simulation in particularly the current density distribution parallel to the wire axis, the associated magnetic field strength, and the temporal evolutions of the peak current density 10 μm away from the laser focus has been added to Fig 2 of the manuscript, making the choice of initial conditions used in the hydro framework more transparent to the reader. In addition we have looked in the bulk heating due to the hot electron circulation also

within the wire. We find a minimal heating of few eV does not change the condition of the shock formation or propagation already 8 μm away from the focal spot as displayed in Fig 2c) of the manuscript. Following referees 2 recommendations we have included more PIC simulations motivating the choice of initial conditions for the MHD simulations and the balance thermal and ablative pressures.

Figure 2: Copper density distribution at 1 ps before arrival of the main pulse.

The PIC extended version of the corresponding section reads now:

To confirm this scenario we have performed 2D PIC simulations of the laser interaction with a 10 μm diameter Cu wire target. We observe the formation of a return peak current of $2.8 \times 10^{13} \text{A/cm}^2$, with a lifetime of 100fs (Fig 2a top and middle show j_y and associated magnetic field for $t=38$ fs after the peak of the interaction, and the time evolution peak of j_y 8 μm away from the focus). The surface return current has two effects on the wire target which can lead to compression: the magnetic compression and Joule heating and the associated ablation. While the return current magnitude scales inverse of the wire radius [42], for a 25 μm diameter copper wire, the peak of the surface return current density is predicted to be in the range of $0.4\text{-}1.1 \times 10^{13} \text{A/cm}^2$, with a current strength decaying further away from the laser focus. At the time when the return current is maximal, the thermal pressure is evaluated to be 8-times higher than the magnetic pressure of 250 Mbar. The resulting plasma $\beta=0.12$ confirms the kinetic nature of our shock formation.

The data presented may show what the authors suggest, but I do not find this single diagnostic output sufficiently compelling, taken alone and without more robust simulation support. High resolution X-UV imaging, x-ray spectroscopy, and other diagnostics, would be well suited to measuring the conditions produced by resistive heating (and possibly the emission from shock waves if the diagnostic is streaked). Such diagnostics have been fielded successfully on previous experiments (including of Cu wire targets of similar dimensions to those used here) by collaborations involving LLNL, OSU, UCSD and others. The single diagnostic approach taken is inadequate to support the claims being made by the authors.

The shock wave we observed is generated far outside the laser focus. That nothing similar was observed in the past although the interaction with similar wires, has been extensively studied, can be explained as the diagnostic tool pool used by previous studies and recommended by the referee to complement our measurements not being suited to observe either the transient surface heating, or the subsequent shock forming at the wire surface, as these diagnostic lack the spatial and temporal resolution, and also the strong background signals generated at the laser focus will hinder the observation of this surface heating. Any emission (thermal or characteristic) by the surface heating is promptly following the emission generated near the laser focus, it will be outshone by it, and cannot be distinguished even by streaking capabilities, also due to the lower yield. Late time information from optical probing is limited to the under-critical plasma regions, and can directly probe just the reflected outward radially moving shock and associated plasma expansion as demonstrated by Yang et al. (Yang 2024). Direct imaging of the full temporal evolution was possible just in the experiment reported in this manuscript as highly relativistic laser matter interaction was paired for the first time with hard x-ray imaging capabilities.

There are features in the radiographs that are not well explained by their hypothesis- for instance at 500ps there appears to be a doubling of the brighter structure on the RHS of the image. If we are to believe that such features are always shock waves, then where does the second shock on the RHS come from? A sudden surface heating should produce just one shock (visible on either side of the target), and a simple homogenous cylinder offers no boundary that could lead to break-up of the primary shock. If such features are not necessarily shock waves, how can we be confident that other features have been identified correctly as shock waves? It is interesting data, but in my view substantially more work is needed - both on the experimental and simulation fronts, to make a compelling case for publication.

We agree with the reviewers comment, a sudden surface heating should produce just one shock and thus, a single peaked structure would appear in a radiograph.

However, the imaging setup used in this experiment results in diffraction effects during the x-ray propagation. Therefore the intensity variation measured by the detector is not only due to absorption effects, as it would be in a radiograph, but includes additional phase propagation terms. Thus, to properly identify the features observed in our experiment, one needs to calculate the x-ray wave-front propagation after the sample up to the detector to include the phase effect. In Figure 3a we show the projected areal mass density along the x-ray propagation direction obtained by our simulation for a set of temporal delays. Then we calculate the x-ray pattern at the back plane of the target as in Fig. 3b. In this case, the reviewer expectations match the calculated x-ray pattern (a single bump, otherwise smooth profile). On Fig 3c, we show the x-ray intensity at the detector plane for the equivalent propagation distance of 6 mm in our

experiment. One can see how the diffraction effects manifest in fringe structures in the data.

Fig 3: a) projected areal mass along the x-ray propagation direction. b) Intensity at contact plane and c) after 6 mm of propagation, as absorption contrast leading to a pure radiograph changes to PCI.

Even then though, the idea of launching shock waves with short pulse lasers is not new - even in a cylindrical geometry. Indeed similar schemes have been proposed previously (for instance by Y. Sentoku) as a potential source of neutrons in heterogenous cylindrical targets imploded by surface resistive heating.

As a clear reference was not provided, we assume a comparison to the (Sentoku 2007) paper. As we have discussed in the overview of the literature in point 2) this reference reports on shocks launched in a cylindrical geometry as a source of neutrons, but with a fundamental difference. The shocks proposed by Sentoku are not cylindrical converging shock, but radially outward oriented, and are hence fundamentally different to the physical setup reported by our work. Also the resistive heating reported by Sentoku is in the bulk (of layered materials) and not limited at the surface.

As regards applications - a convergent geometry is wholly unsuitable for making precise measurements of EOS. It does not offer the necessary diagnostic access and shock wave propagation is complicated by the convergent geometry. EOS experiments are almost always performed in planar geometry with a drive that is as spatially and temporally

uniform as possible. The fact that you can launch a strong shock by some means in no way implies that you have developed a suitable basis for doing EOS experiments.

Here we kindly disagree with the statement of the referee, as shown by substantial literature, converging geometries are a workhorse for EOS at extreme conditions besides the citation 6-11, 53 from the manuscript please also refer in addition to (Absolute Hugoniot measurements from a spherically convergent shock using x-ray radiography , DOI/10.1063/1.5032142).

We have to note that in general also EOS is determined by a single diagnostic, typically VISAR for planar targets. While VISAR in itself also delivers finally single parameter (shock speed) this parameter is related to pressure by using an EOS by making an assumption for the temperature (normally neglected). Our "single" diagnostic is capable to access both the density (on a single shot) and a differential velocity when combining several time delays, thus the pressure can be calculated by the Rankine-Hugoniot relations. The method relies on high reproducibility of the system from shot to shot. This we have proven by shooting the same type of wires on different days with the same delays. For example, in the Figure below we show the 2d intensity measured two weeks after the experiment:

Figure 4: Data from 2 runs at the same experimental conditions, demonstrating the reproducibility of the compression.

Furthermore in recent times experiments exploiting XFEL beams to determine the density in the shock from the atomic lattice distance by high precision diffraction techniques are commonly used (DOI/10.1063/5.0201702). Such method could be applied also for our platform, by focusing the XFEL beam into the shock front. This diagnostic technique is unique to an XFEL due to the short pulse available, and cannot be realized elsewhere.

Conclusion:

It is apparent from the Referee's criticisms that we were perhaps not clear enough in our paper to differentiate between the usual laser-driven shock/blast wave and our new observation of the cylindrical return-current-heating driven shock far removed from the laser focal region. We have changed the narrative to make sure that such a misunderstanding should be avoided for the reader and have added relevant citations. We respectfully disagree with the Referee regarding the necessary suite of shock diagnostics, and have tried to show the superior information that can be obtained with hard x-ray FEL imaging, when properly accounting for diffractive effects. For a more rigorous simulation methodology we have included PIC simulation of a Cu wire to make the choice of initial conditions for the subsequent hydro simulations more transparent.

We hereby are convinced that our arguments have cleared up, the pertinent criticism of the referee, and is now better understandable also for the broad reader.

(Budil 2000) K. S. Budil et al., *Astrophys. J. Suppl. Ser.* 127 262 5 (2000)

(Akli 2008) K. Akli et al. *PRL* 100, 165002 (2008)

(Mondal 2010) S. Mondal et al. *Phys. Rev. Lett.* 105, 105002 (2010)

(Lancaster 2017) K. Lancaster et al. *Physics of Plasmas* 24, 083115 (2017)

(Kamalesh 2018) J. Kamalesh et al. *Phys. Plasmas* 25, 013102 (2018)

(Santos 2017) J. J. Santos et al *New J. Phys.* 19 103005 (2017)

(Sentoku 2007) Y. Sentoku et al. *Phys. Plasmas* 14, 122701 (2007)

(Robinson 2013) A.P.L. Robinson et al. *Phys. Plasmas* 20, 122701 (2013)

(Green 2007) J. Green *Nature Physics* volume 3, pages 853–856 (2007)

Reviewer 2 (Remarks to the Author)

The manuscript presents experimental evidence of dynamic convergent shock compression in thin metallic wires driven by the interaction of tens-of-femtoseconds-duration, Joule-energy level, relativistic-intensity single laser pulse. The results were obtained by adding the radiography probing capability of an XFEL 8.2 keV x-ray source, producing phase contrast images (PCI) of the driven wires at different times.

While this is not the first work providing evidence that short-pulse intense laser pulses can launch strong shocks in solid-density targets [e.g. Budil et al., Astrophys. J. Suppl. Ser. 127 262 5 (2000) (a missing seminal reference, using a low-repetitionrate PW laser system); Santos et al., New J. Phys. 19 (2017) 103005 (should be mentioned that they also used a high-repetition -rate Joule-level laser driver)] and, as such, works as an interesting extension to ns-laser drivers, not yet broadly explored capability to creating warm and hot dense matter states of relevance for studies in planetology and/or astrophysics the results presented here are unprecedented: For the first time it is clear shown such shock propagation inside the dense targets, and this with excellent temporal and spatial resolutions that were unreachable in previous studies using exclusively optic-laser sources.

The formation of the converging shock by a transient energy deposition over a sub- μm thin layer all-around the wires, and this at tens of μm height-offset in respect to the interaction laser axis, is convincing as the experimental features seen in the phase contrast images (shock velocity, transmission radial profiles at selected times) are fairly well reproduced by post-processed 1D cylindrical symmetric hydrodynamic simulations.

The data obtained is of interest and useful for specialists in the fields of high-power laser interaction with matter and extreme states of matter, and as claimed by authors does have the potential for studies relevant to conditions found in Jovian planets or in stars, that is a basis of knowledge development in the broad field of high energy density physics.

We thank the referee for the positive and constructive feedback, and recognition of the relevance of our work. Also in similar to the wishes of referee 1, we have added more historic references to papers reporting on the formation of the shock wave propagating from the laser focal spot. We have also underlined that while this shock and blast wave is observed in our experiment, it is not the focus of our investigation. The corresponding paragraph reads now:

The interaction of such lasers with the matter generates a blast wave following the strong localized heating in the focal spot of the laser [Budil, Modal, Akli, Lancaster, Jana, Santos], and secondary radiation absorbed heterogeneously by the sample can drive hydrodynamic motion [Sentoku, Green, Robinson]. While these past experimental studies have focused in the rarefaction

subsequent to the shock propagation following this blast wave, we experimentally demonstrate that by irradiating a thin wire with a short-pulse laser, conditions are met where a cylindrical shock is generated at the surface that propagates towards the wire axis.

This said, in my opinion, the present manuscript does not sufficiently detail quantitatively the transient, kinetic processes by which the high-intensity laser generates a surface current of hot electrons around the cylinder and the opposite, quasi instantaneously generated, return current of thermal electrons, and that it is the energy deposition by the resistive current transport that drives the ablation of the laser surface and subsequent shock formation. The process is sufficiently described qualitatively providing the needed references from the vast literature on the subject, but the numbers (surface current densities of 10^{13} A/cm², surface heating to electron temperatures above 300 eV) come from unpublished PIC simulation material (cited arXiv reference [36]) that does not study the Cu wires used in this experiment and its exact configuration.

We have corrected the arXiv reference with the published article reference. We have to note the final version of ref [36] includes also data from PIC simulations with a 10 μ m copper wire.

It is the opinion of this referee that the presented PCI data and the physics that it unveils are of the highest standards and deserved to be published, but to meet the publication standards of Nature Communications, the manuscript should provide a comprehensive description of the full system, encompassing both its kinetic and hydrodynamic aspects. In other words, the referee suggests that the authors add results from PIC simulations reproducing the interaction of the high-intensity laser with a 25- μ m-diameter Cu wire, and demonstrate the range of reached surface currents, magnetic-field and transient heating of the material that, ultimately, can allow to discuss on more solid grounds the energy source for driving the observed dynamic radial-shock compression.

Following your recommendations we have analyzed further PIC simulations with Cu as targets, supplementing the claims from the now published article [36]. In particular the current density distribution parallel to the wire axis, the associated magnetic field strength, and the temporal evolution of the peak current density 10 μ m away from the laser focus has been added to Fig 2 of the manuscript. This conditions translate in a surface heating of up to 300 eV used further as initial condition for the hydro simulations, when scaling the PIC results using the theory presented in [36]. We hope the referee agrees with our additions making the choice of initial conditions used in the hydro framework more transparent to the reader.

The corresponding magnetic and thermal pressure have been evaluated at time when the surface current peaks to be 25 TPa and 200 TPa respectively. The resulting plasma $\beta=0.12$ confirms the kinetic nature of our shock formation, opposite to magnetic Z-pinching.

The PIC extended version of the corresponding section reads now:

To confirm this scenario we have performed 2D PIC simulations of the laser interaction with a 10 μm diameter Cu wire target. We observe the formation of a return peak current of $2.8 \times 10^{13} \text{A/cm}^2$, with a lifetime of 100fs (Fig 2a top and middle show j_y and associated magnetic field for $t=38$ fs after the peak of the interaction, and the time evolution peak of j_y , 8 μm away from the focus). The surface return current has two effects on the wire target which can lead to compression: the magnetic compression and Joule heating and the associated ablation. While the return current magnitude scales inverse of the wire radius [42], for a 25 μm diameter copper wire, the peak of the surface return current density is predicted to be in the range of $0.4\text{-}1.1 \times 10^{13} \text{A/cm}^2$, with a current strength decaying further away from the laser focus. At the time when the return current is maximal, the thermal pressure is evaluated to be 8-times higher than the magnetic pressure of 250 Mbar. The resulting plasma $\beta=0.12$ confirms the kinetic nature of our shock formation.

The PIC setup is detailed in the Supplemental Materials:

We address in following your minor questions:

Minor questions: - Should't the colorbar scale for transmission in Fig. 1 b) be limited to a maximum of value of 1?

The imaging conditions established by our setup imply that the intensity measured by the detector results from both absorption and the phase of the propagated x-rays. Therefore the intensity can exceed unity due diffraction effects caused by the propagation distance. This can be easily visualized by using the Transport of Intensity Equation, TIE, to propagate the x-rays over different distances.

At the plane at the back surface of the target (what is usually known as "contact plane") the x-ray propagation distance is 0, and therefore the intensity at this plane is only due to absorption effects. As the x-rays propagate to larger distances, diffraction fringes appear. An example for a cold 25um Cu wire is shown in the Figure 5:

Figure 5: Calculated PCI signal displaying the enhancement of the fringe contrast with larger propagation distance.

- Why is the transmission profile data also analysed at 700 ps, when the maximum compression is reached? Why the corresponding data does not appear if the plot of Fig. 4? This would give a more robust ground to claim pressures in the range of Gbar.

We completely agree with the referee, but as we did not have a simultaneous velocity and density measurement during this experiment, we have refrained to make an experimental claim for the measured pressure.

We have analyzed the 700 ps data in the same manner as the 300 ps and 500 ps data points. The density extracted from the analysis is 104 ± 21 g/cc. Indicating that, as indicated by the simulations, we achieve a large compression at the convergence point. However, we still do not have an experimental velocity for that point and cannot provide a pressure at this point. We have added the reconstructed density value for convergence, and have updated Figure 4 of the manuscript to highlight the density range.

Other minor suggestions:

- Adding the spatial scale to Fig. 1 c).

The scale was added.

-In the literature, the symbol j_h is commonly used for the direct laser-accelerated current-density of hot electrons, not for the return current of thermal electrons for which it is commonly preferred j_e or j_r .

We have implemented the changes now using j_e .

REVIEWER COMMENTS

Reviewer #1 (Remarks to the Author):

I am generally satisfied with the work that the authors have done to improve the manuscript. However, they are incorrect in regard to the work done by Sentoku san, and the novelty of their scheme. Looking at the paper that the authors now propose to cite, you can see the following statement: "Compression factors could be further enhanced by making use of converging shocks, i.e., by embedding cylindrical or spherical metal shells in CD plastic." If I recall correctly, Sentoku expanded on this statement in his talk at the 2006 APS meeting "Shock waves in solids driven by ultra-fast laser heating" Y. Sentoku et al. Before claiming aspects of novelty, I think the authors should clarify exactly what was presented there. I do not want to make too many definitive statements based on a talk that I saw 17.5 years ago. I think it fair to say though that the presentation of an idea/results in an APS DPP meeting talk is a recognised way of establishing precedence in our community. (I do of course recognise that it can be challenging for anyone not present to be aware of what was shown given that only the abstracts are made readily available.)

Reviewer #2 (Remarks to the Author):

This referee considers that the authors have satisfactorily addressed the criticism and comments of both referees regarding their previous submission.

I support publication in Nature Communications with only a minor correction and a few suggestions for improvement:

- The plasma beta parameter is commonly defined as the ratio of thermal pressure over magnetic pressure (and not its inverse): therefore, the value of beta is 8 (instead of 0.12).
- Caption of Fig. 2 – Improve clarity on the origin of the 6 panels: a), b) and c) correspond to the 2D PIC simulations; d), e) and f) to the 1D hydro simulations.
- Also improve the text when introducing the hydro simulations: make a clear transition from PIC to hydro simulations.

Cylindrical compression of thin wires by irradiation with a Joule-class short pulse laser

Reviewer 1 (Remarks to the Author)

I am generally satisfied with the work that the authors have done to improve the manuscript. However, they are incorrect in regard to the work done by Sentoku san, and the novelty of their scheme. Looking at the paper that the authors now propose to cite, you can see the following statement: "Compression factors could be further enhanced by making use of converging shocks, i.e., by embedding cylindrical or spherical metal shells in CD plastic." If I recall correctly, Sentoku expanded on this statement in his talk at the 2006 APS meeting "Shock waves in solids driven by ultra-fast laser heating" Y. Sentoku et al. Before claiming aspects of novelty, I think the authors should clarify exactly what was presented there. I do not want to make too many definitive statements based on a talk that I saw 17.5 years ago. I think it fair to say though that the presentation of an idea/results in an APS DPP meeting talk is a recognised way of establishing precedence in our community. (I do of course recognise that it can be challenging for anyone not present to be aware of what was shown given that only the abstracts are made readily available.)

We thank the referee for the recognition of our efforts to improve the manuscript. Regarding the remaining criticism on the novelty of the scheme driving the shocks in our paper we have followed your advise and have directly contacted Sentoku san, and asked him for a statement, in particularly if his prior work is represented and cited correctly, by providing him both the manuscript and referees 1 comment.

We are happy to report that the novelty of our experimentally demonstrated compression scheme remains untouched, and attach in verbatim Sentoku san's reply:

Dear the reviewer, (From Y. Sentoku)

Thanks for remembering my old work (18 years ago). It is my pleasure that I could impress you my work well. I was asked by the authors to review the manuscript and your comment. Here is what I think from that. In the APS-2006 meeting, I actually did propose a shock compression by heating heterogeneous materials isochorically and drive an isothermal shock to heat ions inside the solid. This work was published in PoP 2007 which is cited in the manuscript.

The cylindrical compression achieved by authors using a short pulse laser was initiated by the surface heating with the return currents which are driven by the fast electrons. So I see that the shock driving mechanism differs from what I proposed in 2006.

Regarding to your concern related to the "converging shocks" by embedding micro shells or cylinders in my work, it was not included both in APS-2006 and PoP 2007 paper. Probably, I discussed in some other conferences around that year, but I feel it is not fair for the authors to ask to cite it since it was not in publications.

From the third person's viewpoint, this paper is impressive, capturing the dynamics of short-pulse driven implosion beautifully by XFEL. I hope that my comment help you reconsider this manuscript.

Best regards,
Yasuhiko Sentoku, Professor
Institute of Laser Engineering, Osaka University

We hope that this clarification is satisfactory and the paper can now be supported for publication.

Reviewer 2 (Remarks to the Author)

This referee considers that the authors have satisfactorily addressed the criticism and comments of both referees regarding their previous submission.

I support publication in Nature Communications with only a minor correction and a few suggestions for improvement:

- The plasma beta parameter is commonly defined as the ratio of thermal pressure over magnetic pressure (and not its inverse): therefore, the value of beta is 8 (instead of 0.12).

Thank you for spotting this error. It has been corrected.

- Caption of Fig. 2 – Improve clarity on the origin of the 6 panels: a), b) and c) correspond to the 2D PIC simulations; d), e) and f) to the 1D hydro simulations.

We have added as proposed a clarification on the origin of the 6 panels. The caption of fig 2 reads now:

Fig. 2 a), b) and c) correspond to the 2D PIC simulations using a copper 10 μm diameter wire; d), e) and f) correspond to the 1D hydro simulations using the SESAME equation of state: a) snapshot the current density parallel to the wire axis and the associated magnetic field strength 38 fs after peak of the interaction, b) time evolution of peak current density 8 μm away from the laser focus and c) distribution of the current density, magnetic field, bulk temperature for the surface layer for $t=38\text{fs}$. Hydrodynamic temporal evolution of compression factor d), pressure e) and temperature f) for a shock driven by a surface temperature of 250 eV.

- Also improve the text when introducing the hydro simulations: make a clear transition from PIC to hydro simulations.

We have added in the narrative the desired clear transition from PIC to hydro, by detailing the multi-scale approach introduced by L. Yang et al. (2024):

... [41], work that follows a multi-scale approach. PIC simulations are used in a first step to calculate the current density profile along an extended wire. With a model detailed by equation 1 a temperature distribution is evaluated from the current density within the skin depth at the wire surface. Finally this temperature distribution is used as initial conditions for hydrodynamic simulations to predict the long-term shock formation, propagation and density compression

We thank the referee for the his comments and support of the work and hope that with this revision the paper is ready for publication.